# MetAP2 as a Therapeutic Target for Obesity and Type 2 Diabetes: Structural Insights, Mechanistic Roles, and Inhibitor Development

**DOI:** 10.3390/biom14121572

**Published:** 2024-12-10

**Authors:** Dong Oh Moon

**Affiliations:** Department of Biology Education, Daegu University, 201, Daegudae-ro, Gyeongsan-si 38453, Gyeongsangbuk-do, Republic of Korea; domoon@daegu.ac.kr

**Keywords:** T2DM, obesity, MetAP2, inhibitors

## Abstract

Type 2 Diabetes Mellitus (T2DM) and obesity are globally prevalent metabolic disorders characterized by insulin resistance, impaired glucose metabolism, and excessive adiposity. Methionine aminopeptidase 2 (MetAP2), an intracellular metalloprotease, has emerged as a promising therapeutic target due to its critical role in regulating lipid metabolism, energy balance, and protein synthesis. This review provides a comprehensive analysis of MetAP2, including its structural characteristics, catalytic mechanism, and functional roles in the pathophysiology of T2DM and obesity. The unique architecture of MetAP2’s active site and its interactions with substrates are examined to elucidate its enzymatic function. The review also explores the development of MetAP2 inhibitors, focusing on their mechanisms of action, preclinical and clinical findings, and therapeutic potential. Special emphasis is placed on docking studies to analyze the binding interactions of six key inhibitors (fumagillin, TNP-470, beloranib, ZGN-1061, indazole, and pyrazolo[4,3-b]indole) with MetAP2, revealing their structural determinants for efficacy and specificity. These findings underscore the potential of MetAP2 as a therapeutic target and provide valuable insights for the rational design of next-generation inhibitors to address obesity and T2DM.

## 1. Introduction

By 2023, approximately 537 million adults aged between 20 and 79 globally were reported to have diabetes, with T2DM representing the majority of cases, accounting for over 90%. Additionally, the prevalence of obesity worldwide has experienced a nearly threefold increase since 1975, with over 1 billion adults now classified as obese [1,2]. These concerning statistics highlight the critical need for innovative therapeutic approaches to combat these closely linked metabolic disorders.

T2DM and obesity are major global health challenges, characterized by complex metabolic dysfunctions that contribute to a range of complications, including cardiovascular diseases, insulin resistance, and chronic inflammation [3,4,5]. The prevalence of these conditions has been steadily increasing over the past decades, largely driven by lifestyle changes, increased caloric intake, and reduced physical activity. T2DM is marked by impaired glucose metabolism, resulting in hyperglycemia due to insulin resistance and β-cell dysfunction [6,7], while obesity is defined by excessive adiposity that disrupts energy homeostasis [8]. Together, these conditions form a tightly linked pathophysiological axis often referred to as the “diabesity epidemic”, highlighting the urgent need for innovative therapeutic strategies.

MetAP2 has been identified as a compelling target for therapeutic intervention in both T2DM and obesity [9,10,11,12,13]. This intracellular metalloprotease plays a pivotal role in post-translational protein modification by catalyzing the removal of the N-terminal methionine from newly synthesized peptides [14,15]. Beyond its enzymatic role, MetAP2 is implicated in pathways regulating lipid metabolism, angiogenesis, and energy expenditure. The pharmacological inhibition of MetAP2 has demonstrated the ability to reduce body weight, enhance glucose tolerance, and improve lipid profiles in preclinical models, making it an attractive target for managing metabolic disorders. Moreover, several MetAP2 inhibitors, such as fumagillin derivatives and newly developed reversible inhibitors, have demonstrated potential in preclinical and early clinical trials for obesity and T2DM.

This review aims to deliver a comprehensive analysis of MetAP2 and its role in metabolic diseases. The structural characteristics and catalytic mechanism of MetAP2 are first explored, focusing on its unique active site architecture and interactions with substrates. This is followed by a discussion of its functional roles in the pathophysiology of T2DM and obesity, including its regulation of lipid metabolism and insulin resistance. Finally, the development and therapeutic potential of MetAP2 inhibitors are examined, with a particular focus on their mechanisms of action, clinical trial outcomes, and future prospects in treating obesity and T2DM. By integrating structural, mechanistic, and pharmacological insights, this review aims to highlight the potential of MetAP2 as a therapeutic target and guide future research toward the rational design of effective and safe MetAP2 inhibitors.

## 2. Structural Characteristics and Mechanism of MetAP2

MetAP2 is an intracellular metalloprotease that plays a crucial role in post-translational modification by cleaving the N-terminal methionine from nascent polypeptides [16,17]. This initial methionine removal is essential for proper protein maturation and function, as approximately 60% of newly synthesized proteins lose their initiator methionine, particularly when the second residue is small and uncharged, such as Ala, Cys, Gly, Pro, Ser, Thr, or Val [18,19]. MetAP2 is evolutionarily conserved across species and exists alongside its homolog, MetAP1, although the two enzymes function independently and have distinct structural and functional characteristics [20]. MetAP2’s activity is crucial in cellular processes, impacting global protein synthesis and cell cycle regulation.

MetAP2 binds specifically to the ribosomal tunnel exit, positioning itself to cleave the initiator methionine from nascent polypeptides as they emerge during translation. The binding of MetAP2 to the ribosome involves interactions with the large ribosomal subunit, particularly through contacts with the expansion segment ES27L of the rRNA [21]. This interaction is facilitated by the N-terminal extension of MetAP2, which acts as a structural stabilizer, anchoring the enzyme at the ribosomal exit site. Interestingly, the N-terminal extension of MetAP2 is capable of autoproteolytic cleavage, producing a modified form of the enzyme that exhibits altered ribosome-binding properties. Upon cleavage, MetAP2 shifts from its initial position at the tunnel exit toward the A-site of the ribosome, a region closely associated with tRNA binding and peptide elongation [21]. This movement likely allows MetAP2 to interact with nascent peptides at different stages of synthesis, adapting its function to varied positions on the ribosome based on the structural state of the enzyme. These dynamic binding properties indicate that MetAP2’s ribosomal association is finely regulated, allowing it to participate in translation processes with spatial precision. This relocation capability may enable MetAP2 to engage with ribosomal complexes in ways that support its role in modulating protein synthesis, particularly by adjusting its proximity to nascent chains for the efficient removal of the N-terminal methionine.

Human MetAP2 is composed of 478 amino acids and exhibits a unique structural arrangement that supports its catalytic function. The enzyme features a central β-sheet core surrounded by α-helices, creating a stable framework for catalysis. Both MetAP2 and its homolog MetAP1 share a conserved “pita bread” fold in the C-terminal domain [21]. This fold forms a deep cleft, which houses the enzyme’s active site and facilitates catalytic activity [20]. The active site of MetAP2 is located within a deep pocket on the concave face of the central β-sheet, embedded in the cleft formed by the enzyme’s pita bread fold. Surrounding the active site are two pairs of α-helices (α1–α2 and α3–α4), which enhance structural stability and contribute to substrate specificity [22]. A distinctive characteristic of MetAP2 is its approximately 165-residue N-terminal extension and a unique 60-residue helical insertion in the C-terminal domain. Although the N-terminal extension does not participate directly in catalysis, it plays a regulatory role in protein synthesis and cell cycle progression by influencing the phosphorylation states of key proteins, including eIF2 and ERK1/2 [14]. This N-terminal extension and helical subdomain further distinguish MetAP2 from MetAP1, which lacks these features, and are generally absent in most prokaryotic MetAPs [23,24].

The active site of MetAP2 is designed with a unique structural motif characteristic of many metalloenzymes, featuring two cobalt or manganese ions. These metal ions are coordinated by bridging carboxylate ligands and a bridging water or hydroxide molecule, which are critical for stabilizing the catalytic transition state and activating the water molecule necessary for peptide bond cleavage. Examples of other enzymes sharing this motif include hemerythrin (a dioxygen carrier protein), ribonucleotide reductase (a dinuclear non-heme iron protein), leucine aminopeptidase, urease, arginase, and various phosphatases and phosphoesterases [25,26,27]. This shared structural characteristic highlights how MetAP2’s active site architecture is optimized for its specific catalytic function—the cleavage of N-terminal methionine residues from nascent polypeptides.

MetAP2’s active site binds two metal ions, typically cobalt but possibly manganese under physiological conditions [23,28,29]. This dinuclear metal-binding site is essential for catalytic function and is stabilized by conserved residues: Asp251, Asp262, His331, Glu364, and Glu459. Asp262 and Glu459 act as bidentate ligands, coordinating both metal ions, while Asp251, Glu364, and His331 each coordinate one of the metal ions individually [23,24]. This arrangement secures the metal ions within the active site, enabling effective catalysis. Additionally, the active site features a specificity pocket formed by residues such as Tyr444, which stabilizes the binding of the methionine side chain in substrates. The pocket is partially enclosed, creating an optimal environment for substrate interaction, while the positioning of Tyr444 and the reorientation of nearby water molecules allow the pocket to be exposed to the solvent, enhancing the substrate accessibility [14,24].

The catalytic mechanism of MetAP2 is intricate, relying on the precise coordination between metal ions and active site residues to facilitate the cleavage of the N-terminal methionine. In its active form, MetAP2 binds two closely positioned metal ions, typically cobalt or manganese, which are critical for the enzyme’s catalytic efficiency. These metal ions are coordinated by conserved residues, Asp251, Asp262, His331, Glu364, and Glu459, within the active site, and they play distinct roles in the reaction. The positioning of these metal ions creates a dinuclear metal-binding site, which is essential for catalytic function [22].

The two metal ions in the active site enable the activation of a bridging water molecule. This water molecule forms bonds with both metal ions, positioning it centrally within the active site. Upon binding, the water molecule undergoes deprotonation, facilitated by the second metal ion, which stabilizes the transition to a hydroxide ion, making it a potent nucleophile. This reactive hydroxide ion then attacks the carbonyl carbon of the peptide bond adjacent to the N-terminal methionine residue. The resulting nucleophilic attack destabilizes the peptide bond, leading to its cleavage and subsequent release of the methionine [22,30].

During catalysis, each metal ion fulfills a specific role: one metal ion stabilizes the negatively charged transition state that forms as the peptide bond is broken, while the second metal ion promotes deprotonation of the water molecule, effectively enabling it to act as a nucleophile. This cooperative action between the two metal ions not only facilitates bond cleavage but also stabilizes the uncharged amino group of the methionine residue, ensuring proper substrate orientation and optimal conditions for catalysis. The specificity pocket, composed of surrounding β-strands and α-helices, is configured to allow only substrates with an N-terminal methionine to access the active site. Here, Tyr444 plays a crucial role by interacting with the methionine side chain, properly orienting the substrate for efficient catalysis [14].

Inhibitors such as fumagillin take advantage of this catalytic arrangement by forming a covalent bond with His231 in the active site, thereby irreversibly inhibiting MetAP2. Fumagillin’s epoxide ring covalently attaches to the imidazole nitrogen of His231, forming a stable complex that obstructs substrate access and permanently deactivates the enzyme. This covalent binding displaces a water molecule that is typically coordinated with one of the metal ions, further locking the inhibitor in place and stabilizing the active site in an inhibited conformation. The interaction of MetAP2 with the ribosome, the structure of MetAP2, and the active site and catalytic mechanism of MetAP2 are illustrated in Figure 1.

## 3. MetAP2 Expression in Obesity and T2DM

MetAP2 exhibits distinct expression patterns across various tissues and is implicated in metabolic processes related to obesity and T2DM. In lean mice, MetAP2 is highly expressed in metabolically active tissues such as the liver, brain, skeletal muscle, and intestines, while showing relatively low expression in adipose tissues [31]. However, in diet-induced obesity (DIO) models, significant alterations in MetAP2 expression are observed, reflecting its role in metabolic dysregulation.

In the intestines of DIO mice, MetAP2 mRNA levels are consistently elevated across all regions, suggesting an adaptive or pathological response to high-fat diets [31]. In the liver, MetAP2 expression shifts from a zonal distribution, predominantly in zone 3 hepatocytes of lean mice, to a diffuse pattern in DIO mice, accompanied by an increased expression in infiltrating immune cells. These changes in hepatic MetAP2 expression are associated with liver inflammation and fat accumulation, hallmark features of metabolic syndrome. In skeletal muscle, MetAP2 expression is lower than in the liver and brain, but alterations in DIO mice indicate its involvement in muscle-related insulin signaling pathways. Furthermore, in the brain and peripheral neurons, MetAP2’s high expression highlights its potential role in regulating satiety and energy balance, with pharmacological inhibition leading to suppressed food intake and body weight reduction in preclinical studies.

The connection between MetAP2 and metabolic disorders is further supported by studies in adipose tissue. Although MetAP2 expression is relatively low in lean adipose tissue, obesity is associated with an increase in its expression, particularly in white adipose tissue, where it may contribute to lipid accumulation and reduced energy expenditure. This is consistent with findings that the pharmacological inhibition of MetAP2 in obese models leads to significant fat mass reduction, likely through a combination of decreased food intake, enhanced energy expenditure, and the modulation of adipocyte metabolism.

In T2DM, elevated MetAP2 expression has been linked to insulin resistance and impaired glucose metabolism. Increased expression in the liver is associated with hepatic steatosis and disrupted insulin signaling, while in skeletal muscle, it may interfere with glucose uptake and energy utilization. In pancreatic β-cells, dysregulated MetAP2 expression could contribute to impaired insulin secretion, further exacerbating hyperglycemia.

These findings suggest that MetAP2 serves as a critical node linking metabolic processes in obesity and T2DM, influencing both systemic and cellular pathways. Understanding the tissue-specific roles of MetAP2 and its alterations in metabolic diseases offers insights into potential therapeutic strategies targeting this enzyme to alleviate obesity, improve glucose homeostasis, and mitigate complications associated with T2DM.

## 4. Function of MetAP2 in T2DM and Obesity

The exact mechanism behind the anti-obesity and anti-T2DM effects of MetAP2 remains unclear. However, it is believed that MetAP2’s non-enzymatic role in suppressing the activity of ERK1/2 plays a significant part in this effect [32]. Inhibiting MetAP2 reduces the activity of SREBP (sterol regulatory element-binding protein), which, in turn, lowers lipid and cholesterol synthesis through ERK-related pathways [33,34].

Protein N-myristoylation is a crucial lipid modification involving the attachment of a myristoyl group to a protein’s N-terminal glycine residue. This modification is essential for protein stability, membrane association, and intracellular localization [35,36]. MetAP2 initiates N-myristoylation by cleaving the N-terminal methionine from nascent polypeptides, thereby exposing a glycine residue required for subsequent lipid modification [37]. Once the methionine is removed, N-myristoyltransferase (NMT) catalyzes the transfer of myristic acid from myristoyl-CoA to the glycine residue [38,39].

Certain proteins requiring MetAP2-dependent N-myristoylation—including Gαi, PKCε, and TRAM (TRIF-related adaptor molecule)—play significant roles in metabolic regulation [40,41]. For instance, Gαi, which inhibits adenylate cyclase to reduce intracellular cAMP, relies on N-myristoylation for membrane anchoring and stability [42,43,44]. Studies indicate that the incubation of HEPG2 cells with myristic acid induces the N-myristoylation of PKCε, leading to its constitutive phosphorylation at Thr566/Ser729 within the kinase domain, which is essential for PKCε activity. Moreover, TRAM’s role in TLR4 signal transduction also depends on myristoylation, which enables TRAM to colocalize with TLR4 at the plasma membrane [45].

While MetAP2 inhibitors show promise in treating T2DM and obesity, and while Gαi, PKCε, and TRAM significantly impact these conditions, no direct studies have linked MetAP2 inhibition to the modulation of these proteins to alleviate T2DM or obesity. This review synthesizes existing research to suggest possible signaling pathways through which MetAP2 inhibition might beneficially affect these metabolic disorders.

In adipose tissue, MetAP2 inhibition may elevate cAMP levels by inhibiting Gαi, which typically acts to suppress cAMP production by inhibiting adenylate cyclase. The rise in cAMP subsequently activates PKA, which in turn phosphorylates and activates lipolytic enzymes such as hormone-sensitive lipase (HSL), enhancing lipid breakdown [46,47]. This cascade not only promotes lipolysis but also activates CREB, which induces the expression of thermogenic genes, including Uncoupling Protein 1 (UCP1), in brown and beige adipocytes. The increased UCP1 expression supports mitochondrial proton leak, driving thermogenesis and increasing energy expenditure, which helps reduce fat accumulation and addresses obesity [48,49]. Additionally, the elevated cAMP levels from Gαi inhibition enhance insulin sensitivity by activating PKA-dependent signaling, facilitating GLUT4 translocation in skeletal muscle, and promoting insulin-independent glucose uptake. This mechanism is particularly advantageous in insulin-resistant states such as T2DM, where improved glucose uptake supports glycemic control [50]. Together, these effects contribute to blood glucose regulation and metabolic stability.

Myristoylated PKCε plays a key role in mediating insulin resistance in the liver during lipid overload. Elevated plasma fatty acids from a high-fat diet or overnutrition lead to diacylglycerol (DAG) accumulation in liver cells, which activates PKCε. Myristoylation anchors PKCε to the cell membrane, enhancing its stability and activity. Once activated, PKCε inhibits IRS-1/IRS-2 phosphorylation, disrupting PI3K activation and downstream insulin signaling [51,52]. This disruption decreases glycogen synthesis and promotes gluconeogenesis, increasing blood glucose levels and contributing to hepatic insulin resistance.

A high-fat diet can also stimulate LPS release from the gut [45,53], and the myristoylation of TRAM is essential for regulating TLR4-mediated inflammatory responses in macrophages. When TLR4 binds to LPS, it dimerizes, allowing TRAM to colocalize at the plasma membrane through its N-terminal myristoyl group [45]. Myristoylation is vital for TRAM’s membrane association and interaction with TLR4. Upon LPS stimulation, PKCε phosphorylates TRAM at Ser-16, causing it to dissociate from the membrane. This dissociation enables TRAM to interact with TRIF, activating the NF-κB and IRF3 pathways, which are critical for the inflammatory response [54,55]. Through this pathway, TLR4 activation by LPS increases TNF-alpha expression, a pro-inflammatory cytokine. Secreted TNF-alpha exacerbates insulin resistance by activating JNK, which disrupts insulin signaling in tissues like the liver and skeletal muscle, leading to a decreased insulin sensitivity [56]. Figure 2 illustrates the model of MetAP2-dependent N-myristoylation and its involvement in T2DM and obesity.

The role of MetAP2-dependent N-myristoylation in regulating metabolic pathways provides a promising avenue for managing metabolic diseases. Proteins such as Gαi, PKCε, and TRAM, which depend on N-myristoylation for their function, are implicated in both metabolic regulation and inflammation. Although the therapeutic potential of MetAP2 inhibitors in T2DM and obesity has been recognized, direct links between MetAP2 inhibition and the modulation of these myristoylated proteins remain unproven. Future research should focus on exploring how MetAP2 inhibition affects these signaling pathways to substantiate its role as a therapeutic target for metabolic disorders.

## 5. Types and Roles of MetAP2 Inhibitors

MetAP2 inhibitors have emerged as a significant class of compounds targeting a variety of diseases, including cancer, angiogenesis-related conditions, immune modulation, inflammation, and metabolic disorders such as obesity and T2DM. These inhibitors are broadly classified into two categories: covalent irreversible inhibitors and reversible inhibitors, each leveraging unique mechanisms to block the enzymatic activity of MetAP2.

The development of MetAP2 inhibitors began with the discovery of natural products such as fumagillin and ovalicin, followed by the creation of semi-synthetic analogs like TNP-470 and PPI-2458, which exhibit strong anti-angiogenic properties. Over time, research transitioned toward drug-like molecules with improved pharmacokinetic profiles, such as the reversible inhibitors LAF389 and M8891, which provided enhanced specificity and bioavailability. Building on this foundation, this review aims to focus on six representative MetAP2 inhibitors with preclinical or clinical research outcomes for type 2 diabetes and obesity, exploring their mechanisms of action, therapeutic potential, and results from these studies.

### 5.1. Fumagillin

Fumagillin was first isolated from Aspergillus fumigatus in 1949, and its role as an inhibitor of MetAP2 was subsequently identified in 1997 [57,58]. Shenping Liu et al. provided key insights into fumagillin’s selective inhibition of human MetAP-2 through high-resolution crystallography (1.8 Å), demonstrating that fumagillin forms an irreversible covalent bond with His231 in the MetAP-2 active site. This finding has contributed significantly to structure-based drug design efforts aimed at developing selective and potent MetAP2 inhibitors [22].

The anti-obesity potential of fumagillin has been extensively studied in preclinical models. In 2010, Lijnen et al. demonstrated that fumagillin significantly reduced body weight, gonadal (GON), and subcutaneous (SC) fat masses in high-fat diet-fed 11-week-old C57BL/6J mice treated with 1 mg/kg/day for 4 weeks [57]. The histological analysis revealed smaller, denser adipocytes in treated mice, with fewer blood vessels surrounding adipocytes in GON fat, suggesting an angiogenesis-related mechanism. Furthermore, fumagillin improved the lipid profiles, including reductions in cholesterol, triglycerides, and leptin levels. In a separate study, 20-week-old obese mice treated for 4 days showed a reduced adipocyte size and increased density in SC fat pads. Notably, the treatment upregulated the expression of angiogenesis-related genes, including MetAP2, TIE-2, angiopoietin-1, and angiopoietin-2 in GON adipose tissue, further emphasizing its role in angiogenic pathways.

In 2015, Harman Kalsi and Ravneet K. Grewal investigated the interaction of fumagillin with mouse intestinal P-glycoprotein (Pgp), a key efflux transporter that limits the bioavailability of orally administered drugs [59]. Their findings showed that fumagillin exhibits a strong interaction energy with Pgp, suggesting its potential to inhibit Pgp-mediated drug efflux. This property could enhance the bioavailability and efficacy of oral antidiabetic drugs, positioning fumagillin as a promising adjunct in T2DM management.

Fumagillin’s effects on metabolic parameters were further explored in 2020 by Moore et al. in dogs fed a high-fat and high-fructose diet (HFFD) [60]. An 8-week treatment regimen resulted in a significant reduction in food intake (~29%) and a modest body weight decrease (~5.6%). Notably, oral glucose tolerance tests (OGTT) demonstrated a 44% reduction in glycemic excursions, indicative of an improved glucose tolerance. An enhanced hepatic glucose uptake (NHGU) and increased liver glycogen storage were also observed, highlighting fumagillin’s potential to improve liver glucose metabolism. These results suggest that fumagillin holds promise as a therapeutic agent for addressing glucose intolerance, hepatic glucose uptake deficits, and lipid metabolism disorders in conditions such as prediabetes and T2DM.

Although fumagillin has shown promising preclinical results, its clinical application is limited by potential side effects, including toxicity concerns related to its irreversible inhibition of MetAP2 and impact on angiogenesis. These safety issues have posed challenges for its development as a therapeutic agent. To address these limitations, efforts have focused on the development of fumagillin derivatives with improved safety profiles and targeted action. Compounds such as TNP-470 and newer analogs have been designed to reduce the systemic toxicity while retaining the anti-obesity and anti-diabetic efficacy, offering a more viable path for therapeutic use.

### 5.2. TNP-470

TNP-470, also known as AGM-1470, is a synthetic analog of fumagillin that was developed in 1998 to enhance the potency and selectivity through modifications to the C6 side chain [61]. Specifically, the replacement of fumagillin’s long alkyl ester group with a carbamate group containing a chloroacetyl moiety allows TNP-470 to form a covalent bond with the His231 residue in the active site of MetAP-2 [22]. This structural modification enhances its pharmacological properties, including strong anti-angiogenic and endothelial cell growth-inhibitory effects, making it a more potent therapeutic agent than fumagillin [62].

In a key preclinical study, Rupnick et al. (2002) demonstrated TNP-470’s anti-obesity potential through subcutaneous administration (2.5–10 mg/kg/day) in ob/ob mice [63]. The treatment led to a dose-dependent reduction in body weight gain. Notably, weight loss was reversible when the treatment was paused, with mice regaining weight during off-treatment periods. By the fourth treatment cycle, TNP-470-treated mice maintained lean weights similar to age-matched controls, while untreated mice became severely obese. The treatment also reduced fat mass and induced a metabolic shift toward fat utilization, as indicated by increased basal metabolic rates and a decreased respiratory exchange ratio. These findings suggest that angiogenesis inhibition may regulate energy balance and metabolic efficiency.

White et al. (2012) further evaluated TNP-470’s effects in high-fat diet (HFD)-fed mice, showing significant reductions in body weight gain, epididymal fat, liver weight, and hepatic lipid accumulation [64]. By day 5, the TNP-470-treated mice consumed fewer calories, and by day 15, their caloric intake aligned with chow-fed controls despite continued access to an HFD. Importantly, the TNP-470-treated mice exhibited an increased energy expenditure over 24 h, independent of locomotor activity. The authors proposed a novel vascular signaling mechanism in adipose tissue that could centrally regulate energy balance, linking TNP-470’s angiogenic inhibition to metabolic enhancements.

Craig et al. (2021) highlighted TNP-470’s role in T2DM management by investigating its combination with sitagliptin in high-fat-fed diabetic mice [65]. TNP-470 alone reduced blood glucose, improved glucose tolerance, and increased xenin levels, a peptide hormone associated with metabolic regulation. When combined with sitagliptin, the treatment produced synergistic effects, including faster glucose normalization, superior glucose tolerance, and enhanced pancreatic beta-to-alpha cell ratios. These findings position TNP-470 as a potential enhancer of incretin-based T2DM therapies.

These studies collectively underscore TNP-470’s multifunctional properties in regulating energy balance, enhancing metabolic efficiency, and improving outcomes in obesity and T2DM. Its mechanisms, potentially involving adipose tissue vascular modulation and xenin upregulation, make it a promising candidate for addressing obesity-T2DM comorbidities.

Despite its preclinical promise, TNP-470 has primarily been evaluated in clinical trials for its anti-angiogenic properties in cancer treatment [66,67,68,69]. However, its clinical application is constrained by neurological toxicity. Emerging evidence from preclinical models highlights its potential in metabolic disorders, offering benefits such as reduced adipose vascularization, improved glucose metabolism, and weight reduction. Notably, its combination therapy potential, as demonstrated with sitagliptin, suggests a valuable role in dual-action therapies for obesity and T2DM.

### 5.3. Beloranib (CKD-732, ZGN-440, or ZGN-433)

Beloranib, a synthetic derivative of TNP-470, was developed in 2005 to improve water solubility and pharmacological properties while retaining the fumagillol backbone [70]. Structural modifications to the phenyl ring enhance hydrophobic interactions with key MetAP2 residues, such as Leu323 and Leu447, thereby increasing its binding affinity and potency. Compared to TNP-470, beloranib exhibits superior efficacy in tumor growth suppression, demonstrating therapeutic advantages over its precursor.

Initially designed as an angiogenesis inhibitor for cancer treatment, beloranib was later repurposed for obesity management due to its promising anti-obesity effects. Developed by Chong Kun Dang Pharmaceutical Corp (Seoul, South Korea). and Zafgen Inc (Boston, MA, USA)., it targets MetAP2 to modulate lipid metabolism and energy expenditure, resulting in significant weight loss and metabolic improvements in both preclinical and clinical studies [71].

In preclinical models (2007), beloranib significantly reduced cumulative food intake and body weight, with more pronounced effects in obese subjects. The treated mice exhibited increased core temperatures, indicating elevated energy expenditure. The fat pad weights and adipocyte size were also markedly reduced. Intracerebroventricular injections demonstrated central anorexic effects, implicating hypothalamic involvement in appetite suppression. Additionally, conditioned taste aversion (CTA) was observed, further contributing to reduced food intake, particularly in obese models [71].

Clinical trials confirmed beloranib’s efficacy in humans. A Phase I ascending dose-controlled trial by Hughes et al. (2013) evaluated its safety, tolerability, and weight-loss efficacy in obese women [72]. Over four weeks, participants receiving intravenous doses of 0.1, 0.3, or 0.9 mg/m^2^ experienced dose-dependent weight loss, with the highest dose group losing a median of 3.8 kg versus 0.6 kg in the placebo group. Improvements in lipid profiles, including a 42% reduction in triglycerides and an 18% reduction in LDL cholesterol, were noted. Adverse events such as nausea and infusion site injury were mild to moderate.

Kim et al. (2015) conducted a 12-week Phase II randomized, double-blind study involving 147 obese adults [73]. Subcutaneous doses of 0.6, 1.2, and 2.4 mg resulted in weight losses of 5.5 kg, 6.9 kg, and 10.9 kg, respectively, compared to 0.4 kg in the placebo group. Additional benefits included reductions in waist circumference, body fat mass, triglycerides, and high-sensitivity C-reactive protein. Mild to moderate gastrointestinal and sleep-related adverse events were observed, particularly at higher doses.

Shoemaker et al. (2017) examined beloranib in patients with hypothalamic injury-associated obesity (HIAO) [74]. Over four to eight weeks, beloranib treatment resulted in weight losses of 3.2 kg and 6.2 kg, respectively, with improvements in the high-sensitivity C-reactive protein. Adverse events were mild to moderate, and no participants discontinued treatment.

In a Phase II study on Prader–Willi syndrome (PWS), McCandless et al. (2017) reported significant reductions in hyperphagia-related behaviors and body weight over 26 weeks [75]. Weight losses of 8.2% and 9.5% were observed with 1.8 mg and 2.4 mg doses, respectively. However, the trial was terminated early due to thromboembolic events, including two fatal pulmonary embolisms.

Proietto et al. (2018) investigated beloranib in obese individuals with T2DM [76]. A 26-week study showed weight reductions of 13.5% and 12.7% and significant HbA1c improvements with 1.2 mg and 1.8 mg doses, respectively. The trial was terminated early due to thromboembolic risks.

Although the clinical development of beloranib was halted due to safety concerns, particularly thromboembolic events, its mechanism, targeting MetAP2 to influence lipid metabolism and energy expenditure, highlights its potential as a therapeutic candidate for obesity and related metabolic disorders. Future research may refine its safety profile or lead to the development of safer derivatives.

### 5.4. ZGN-1061

ZGN-1061 is a second-generation MetAP2 inhibitor developed in 2018 to address the safety concerns associated with beloranib, its predecessor. This compound was designed to retain the therapeutic efficacy of MetAP2 inhibition while reducing adverse effects, particularly thromboembolic risks, through pharmacokinetic modifications such as shorter drug exposure times and lower intracellular accumulation [77].

In preclinical studies (2018), ZGN-1061 demonstrated significant anti-obesity and antidiabetic effects. Four weeks of subcutaneous administration in diet-induced obese (DIO) insulin-resistant mice led to a 25% reduction in body weight, primarily due to decreased fat mass. Additionally, ZGN-1061 improved metabolic parameters, including reductions in plasma glucose and insulin levels. These effects were comparable to beloranib but were achieved with markedly reduced adverse impacts on endothelial cell proliferation and coagulation proteins [77].

The improved safety profile of ZGN-1061 was further confirmed in animal studies (2018). In dogs, ZGN-1061 exhibited rapid absorption and clearance, with a shorter half-life compared to beloranib. Unlike its predecessor, ZGN-1061 did not increase coagulation markers in dogs and showed a significantly better safety profile in rats. The shorter duration of exposure and reduced cellular inhibition of MetAP2 are key factors contributing to the enhanced safety of ZGN-1061 [77].

Early clinical evaluations (2018) also provided promising results. A Phase I randomized, double-blind trial investigated single ascending doses (SAD) and multiple ascending doses (MAD) of ZGN-1061 in healthy individuals and subjects with obesity [78]. ZGN-1061 was well tolerated across all doses, with the most common adverse events being a mild headache and procedural irritation. The pharmacokinetic profile showed rapid drug absorption and clearance, consistent with preclinical findings. In the MAD phase, ZGN-1061-treated participants showed trends in weight loss (−1.5 kg for ZGN-1061 vs. −0.2 kg for placebo) and biomarker improvements, indicating preliminary efficacy.

A Phase II trial in 2020 evaluated ZGN-1061 in overweight and obese adults with T2DM [79]. Participants received subcutaneous doses of 0.05, 0.3, 0.9, or 1.8 mg every three days for 12 weeks. Clinically meaningful reductions in HbA1c were observed at 0.9 mg (−0.6%) and 1.8 mg (−1.0%) doses relative to the placebo, accompanied by a weight loss of 2.2% in the 1.8 mg group. Adverse events were balanced across treatment groups, and no safety signals were identified. However, Zafgen later discontinued the development of MetAP2 inhibitors, including ZGN-1061, citing unresolved safety concerns.

Despite its termination, ZGN-1061 demonstrated the potential of MetAP2 inhibition as a therapeutic strategy for obesity and T2DM. The reduced thromboembolic risk and improved metabolic outcomes highlight the promise of next-generation MetAP2 inhibitors with further optimization. Continued research into the molecular mechanisms of MetAP2 and its role in metabolic diseases may lead to safer and more effective treatments.

### 5.5. Indazole

Indazole, a bicyclic heterocycle composed of fused benzene and pyrazole rings, serves as a versatile scaffold in medicinal chemistry due to its ability to interact with diverse biological targets. Compound 38 is an indazole derivative obtained through Structure–Activity Relationship (SAR) studies aimed at optimizing its inhibitory activity against MetAP2.

In the structural design of Compound 38, a trifluoromethyl (CF3) substituent was introduced at position 6 of the indazole scaffold, significantly enhancing its potency and lipophilic ligand efficiency (LLE), while an aromatic or heteroaryl group at position 4 was optimized to fill a hydrophobic cleft near key residues like Tyr444 and His339, improving the binding affinity and metabolic stability [80]. Compound 38 exhibits remarkable pharmacological properties, including low IC50 values (5 nM), indicating a high efficacy against MetAP2, favorable pharmacokinetics with good oral bioavailability (58%) and a half-life of 2 h, and exceptional selectivity, with no significant off-target effects observed in a panel of 100 biological targets.

In vivo, Compound 38 demonstrated robust and sustained dose-dependent weight loss in a DIO mouse model over 28 days, with no notable toxicities, positioning it as a potent, selective, and safe therapeutic candidate for obesity treatment.

### 5.6. Pyrazolo[4,3-b]indole

Pyrazolo[4,3-b]indole is a tricyclic heterocyclic compound formed by the fusion of pyrazole and indole rings, known for its strong binding affinity to biological targets due to its unique structure. It is a versatile scaffold in medicinal chemistry, offering flexibility for structural modifications and diverse substituents. Compound 10, a derivative of Pyrazolo[4,3-b]indole, was designed based on SAR studies to enhance its inhibitory activity against MetAP2.

Compound 10 features a pyrazole warhead where the N2 nitrogen directly interacts with a metal ion in the MetAP2 active site, while the N1 nitrogen forms a water-mediated interaction with another metal ion. At the 4-position, an aryl substituent fills the hydrophobic cleft between Tyr444 and His339, improving binding affinity. Additionally, a fluorine atom at the 7-position optimally fits into the hydrophobic pocket, enhancing potency and metabolic stability [81].

Pharmacologically, Compound 10 exhibits excellent properties, including an oral bioavailability of 70%, a half-life of 6 h, and good stability in human liver microsomes. In vivo studies demonstrated that Compound 10 induces dose-dependent weight loss, with a 4% body weight reduction observed in diet-induced obesity (DIO) mice after 7 days of oral administration at 30 mg/kg. Importantly, the compound showed no significant off-target effects, toxicity, or impact on food intake at this dose.

The effects of MetAP2 inhibitors, their mechanisms of action, and corresponding clinical trial stages are summarized in Table 1, while the Summary of MetAP2 Inhibitors: Binding Analysis, Docking Insights, and Key Results is presented in Table 2.

## 6. Docking Studies of MetAP2 Inhibitors for Obesity and T2DM

MetAP2 inhibitors represent a promising therapeutic class for obesity and T2DM, but as shown in Table 2, docking results for these inhibitors remain unreported. To address this gap, ligand-protein docking studies were conducted using CB-Dock2 (http://183.56.231.194:8001/cb-dock2/index.php, accessed on 14 November 2024) and visualized with the Biovia Discovery Studio Visualizer (version 21.1.0.20298) to investigate the interactions between MetAP2 and six key inhibitors: fumagillin, TNP-470, beloranib, ZGN-1061, indazole (Compound 38), and pyrazolo[4,3-b]indole (Compound 10). The chemical structures of these inhibitors were retrieved from PubChem (https://pubchem.ncbi.nlm.nih.gov/, 14 November 2024). This study aims to predict binding interactions, assess binding affinities, and contribute to understanding the mechanisms of action of these inhibitors in the context of metabolic disorders.

Docking studies are invaluable in drug discovery as they provide structural insights by predicting how ligands bind within the active site of a target protein, revealing key molecular interactions such as hydrogen bonding, hydrophobic interactions, and metal coordination [82,83]. They also estimate binding affinities, offering quantitative predictions of interaction strength, and facilitate ligand optimization by identifying critical binding features for designing more potent inhibitors. Moreover, docking allows for the rapid screening of multiple ligands while enhancing mechanistic understanding by generating hypotheses about inhibitor action that can be validated experimentally.

To investigate MetAP2 inhibitors, the protein structure of MetAP2 was obtained from the Protein Data Bank (PDB ID: 1BOA) and pre-processed to optimize the docking conditions. Chemical structures of the six inhibitors were retrieved from PubChem and energy-minimized for the docking simulations. Using CB-Dock2, each ligand’s binding site and pose were predicted, with key parameters such as the binding energy and interaction types recorded. The docking scores and detailed binding interactions provided key insights into the molecular mechanisms of these inhibitors, highlighting their potential for obesity and T2DM treatment. Figure 3 and Figure 4 illustrate the binding poses and interactions for each inhibitor, along with their associated docking scores and contact residues.

The docking scores ranged from −6.2 kcal/mol for beloranib to −8.3 kcal/mol for indazole, indicating strong binding affinities across all inhibitors. Among the inhibitors, fumagillin (−6.7 kcal/mol) and TNP-470 (−6.8 kcal/mol) formed stable interactions with His231, a critical residue for MetAP2 activity. These interactions were further stabilized by additional hydrogen bonds and hydrophobic contacts with residues such as Tyr444, Leu447, and His382. This confirms their role as covalent inhibitors targeting angiogenesis pathways.

Beloranib and ZGN-1061, designed as second-generation inhibitors, exhibited promising binding scores (−6.2 and −7.2 kcal/mol, respectively) and interactions with key residues, including His231 and His339. These compounds demonstrated effective binding to the MetAP2 active site, occupying the hydrophobic pocket with their unique substituents. Notably, ZGN-1061’s improved safety profile compared to beloranib may stem from its selective binding characteristics and reduced off-target interactions.

Indazole and pyrazolo[4,3-b]indole, as reversible inhibitors, showed the strongest docking scores (−8.3 and −8.1 kcal/mol, respectively). Both inhibitors formed robust interactions with the active site metal ions via their nitrogen-containing warheads. Their hydrophobic substitutions at the 6- and 7-positions effectively filled the adjacent pockets, enhancing binding stability. These results align with their reported efficacy in preclinical models, emphasizing their potential as orally bioavailable MetAP2 inhibitors.

Overall, the docking studies provide valuable structural insights into the interactions between MetAP2 inhibitors and the enzyme’s active site. The data support the hypothesis that both covalent and reversible inhibitors can achieve strong and specific binding to MetAP2. Furthermore, the findings highlight the importance of interactions and coordination with active site residues, such as His231 and His339, for inhibitor potency. The strong interactions observed between the inhibitors and specific residues, such as Tyr444 and Leu447, suggest that modifying substituents to better occupy these hydrophobic pockets could increase the selectivity and reduce off-target effects. This approach may be particularly valuable in differentiating reversible inhibitors like indazole and pyrazolo[4,3-b]indole from covalent inhibitors such as fumagillin, which have higher toxicity profiles. Future research should focus on optimizing these compounds to balance the efficacy and safety, particularly for clinical applications in obesity and T2DM management. By integrating docking data with experimental findings, this study contributes to the rational design of next-generation MetAP2 inhibitors with enhanced therapeutic profiles.

## 7. Conclusions

MetAP2 is a critical enzyme involved in metabolic regulation, making it a promising target for therapeutic intervention in obesity and T2DM. The unique structural and catalytic features of MetAP2, including its dinuclear metal-binding site and substrate specificity, underscore its role in lipid metabolism, energy balance, and glucose homeostasis. This review highlights the significant progress in the development of MetAP2 inhibitors, ranging from natural products like fumagillin to advanced chemotypes such as indazole and pyrazolo[4,3-b]indole. Docking studies provide further mechanistic insights into the interactions between MetAP2 and its inhibitors, emphasizing key binding residues and structural determinants for efficacy.

Despite promising preclinical and early clinical results, challenges remain, including safety concerns and limited clinical success, particularly with earlier generations of inhibitors. The findings presented in this review underscore the importance of optimizing the pharmacokinetic profiles and safety margins of MetAP2 inhibitors to enhance their therapeutic potential. Future research should focus on developing selective, reversible inhibitors with reduced off-target effects and improved metabolic profiles. By integrating structural, mechanistic, and pharmacological insights, this review aims to pave the way for the rational design of next-generation MetAP2 inhibitors, offering new hope for effective and safe treatments for obesity and T2DM.

## Figures and Tables

**Figure 1 biomolecules-14-01572-f001:**
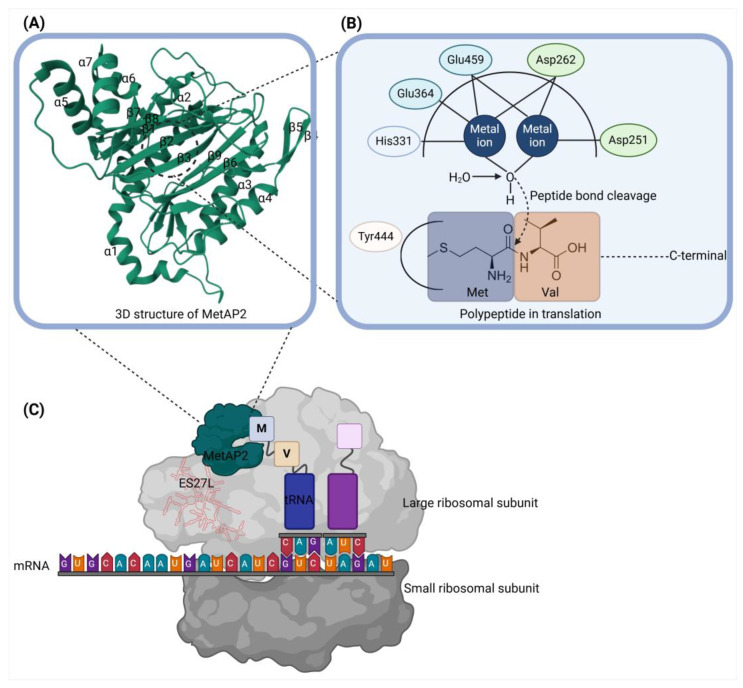
Structure and function of MetAP2 in ribosomal interaction and catalytic mechanism. (**A**) 3D structure of MetAP2. The structure of human MetAP2 (composed of 478 amino acids) features a central β-sheet core surrounded by α-helices (α1–α7), providing structural stability necessary for its catalytic function. The conserved “pita bread” fold within the C-terminal domain forms a deep cleft that houses the active site. The 3D structure of MetAP2 was obtained from the Protein Data Bank (PDB ID: 1BOA, https://www.rcsb.org/structure/1BOA, accessed on 11 November 2024). (**B**) Active site and catalytic mechanism of MetAP2. The active site, located within the deep pocket of the pita bread fold, binds two metal ions, typically cobalt but potentially manganese under physiological conditions. These metal ions are stabilized by conserved residues, including Asp251, Asp262, His331, Glu364, and Glu459. Asp262 and Glu459 act as bidentate ligands, coordinating both metal ions, while Asp251, Glu364, and His331 each coordinate one of the metal ions individually. The metal ions activate a bridging water molecule (H_2_O), converting it into a nucleophilic hydroxide ion (OH^−^) through deprotonation. This hydroxide ion then attacks the carbonyl carbon of the N-terminal methionine peptide bond, leading to peptide bond cleavage and the removal of the initiator methionine from the nascent polypeptide. (**C**) MetAP2 binds to the large ribosomal subunit at the tunnel exit, where it specifically interacts with the expansion segment ES27L of the rRNA. This interaction positions MetAP2 to efficiently engage with the emerging nascent polypeptide, allowing it to cleave the N-terminal methionine as the polypeptide exits the ribosome. The figure illustrates MetAP2’s location relative to the mRNA and tRNA during translation, with ES27L providing an anchor point that stabilizes MetAP2 at the ribosome. This precise positioning enables MetAP2 to facilitate the maturation of newly synthesized proteins by selectively removing the initiator methionine from nascent polypeptides.

**Figure 2 biomolecules-14-01572-f002:**
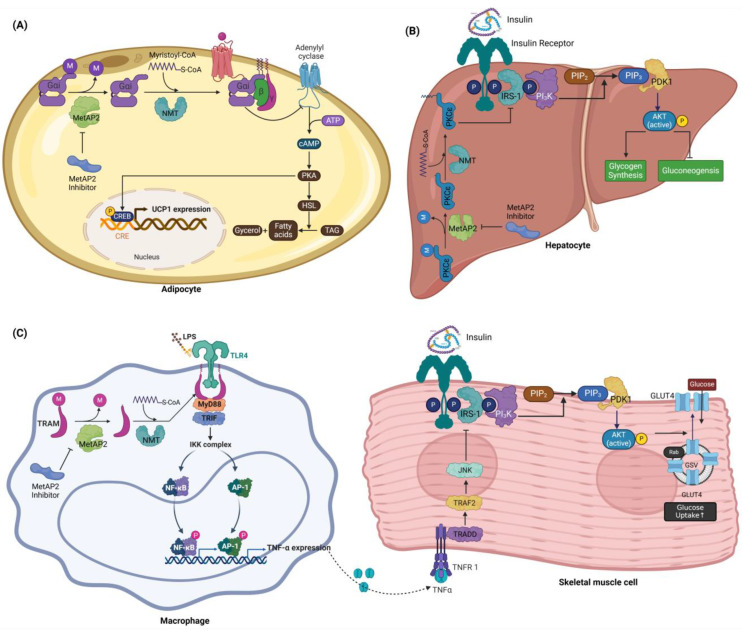
A model of MetAP2-dependent N-myristoylation and its role in T2DM. (**A**) Adipocyte: The model illustrates the effects of MetAP2-dependent N-myristoylation on cAMP signaling and lipid metabolism within adipocytes. The myristoylation of Gαi enables its membrane localization, where it suppresses adenylate cyclase activity, reducing cAMP levels. MetAP2 inhibition prevents Gαi membrane anchoring, resulting in elevated cAMP levels. This activates PKA, which phosphorylates hormone-sensitive lipase (HSL), promoting triglyceride breakdown into glycerol and free fatty acids. Concurrently, cAMP activates CREB, enhancing the transcription of thermogenic genes, such as UCP1, to increase mitochondrial thermogenesis and energy expenditure, reducing fat accumulation and aiding in obesity management. (**B**) Hepatocyte: The panel highlights the role of myristoylated PKCε in promoting hepatic insulin resistance. Elevated fatty acid levels increase diacylglycerol (DAG) accumulation, activating PKCε, which is stabilized on the membrane through myristoylation. Activated PKCε inhibits IRS-1/IRS-2 phosphorylation, impairing PI3K/AKT signaling, reducing glycogen synthesis, and increasing gluconeogenesis. MetAP2 inhibition could modulate PKCε activity by preventing its myristoylation, offering potential therapeutic effects against lipid-induced hepatic insulin resistance. (**C**) Macrophage and skeletal muscle cell: The model illustrates the importance of TRAM’s N-myristoylation in TLR4-mediated inflammatory signaling. Upon LPS binding, TLR4 dimerizes, and myristoylated TRAM colocalizes with TLR4 on the membrane. PKCε phosphorylates TRAM at Ser-16, causing its dissociation from the membrane to interact with TRIF. This activates NF-κB and IRF3 pathways, inducing TNF-alpha expression. Secreted TNF-alpha exacerbates insulin resistance by activating JNK, which disrupts insulin signaling in skeletal muscle cells. TNF-alpha activates JNK, which inhibits IRS-1 phosphorylation and downstream PI3K/AKT signaling. This impairs GLUT4 translocation to the plasma membrane, reducing glucose uptake and contributing to insulin resistance.

**Figure 3 biomolecules-14-01572-f003:**
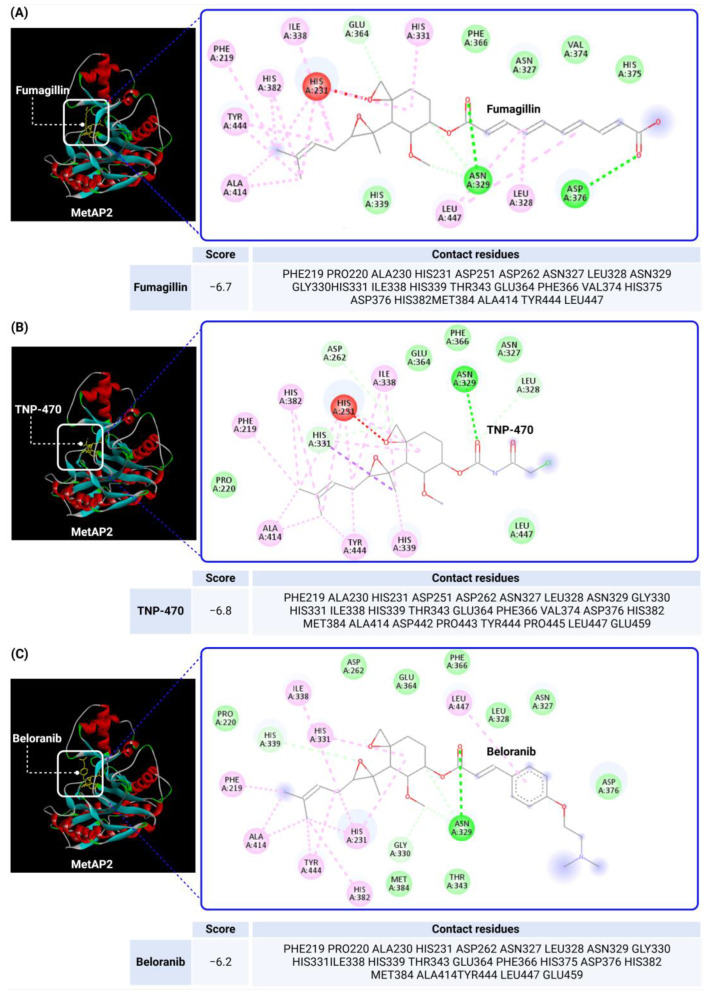
Docking results of Fumagillin, TNP-470, and Beloranib with MetAP2. The docking results of fumagillin (**A**), TNP-470 (**B**), and beloranib (**C**) with MetAP2, performed using CB-Dock2 (https://cadd.labshare.cn/cb-dock2/, accessed on 14 November 2024) and visualized with the Biovia Discovery Studio Visualizer (version 21.1.0.20298), are presented. The MetAP2 structure (PDB ID: 1BOA) was obtained from the Protein Data Bank, and ligand structures (SDF files) were downloaded from PubChem. The docking process was conducted using CB-Dock2, which predicted binding poses and calculated binding scores (in kcal/mol). The binding scores for fumagillin, TNP-470, and beloranib were −6.7, −6.8, and −6.2 kcal/mol, respectively. Key contact residues involved in hydrogen bonding and hydrophobic interactions are displayed in the 2D interaction diagrams, including His231, Tyr444, Leu447, and His382. These residues are critical for stabilizing the inhibitors within the MetAP2 active site. The docking study highlights the covalent and hydrophobic interactions, particularly with His231, which is essential for MetAP2 inhibition.

**Figure 4 biomolecules-14-01572-f004:**
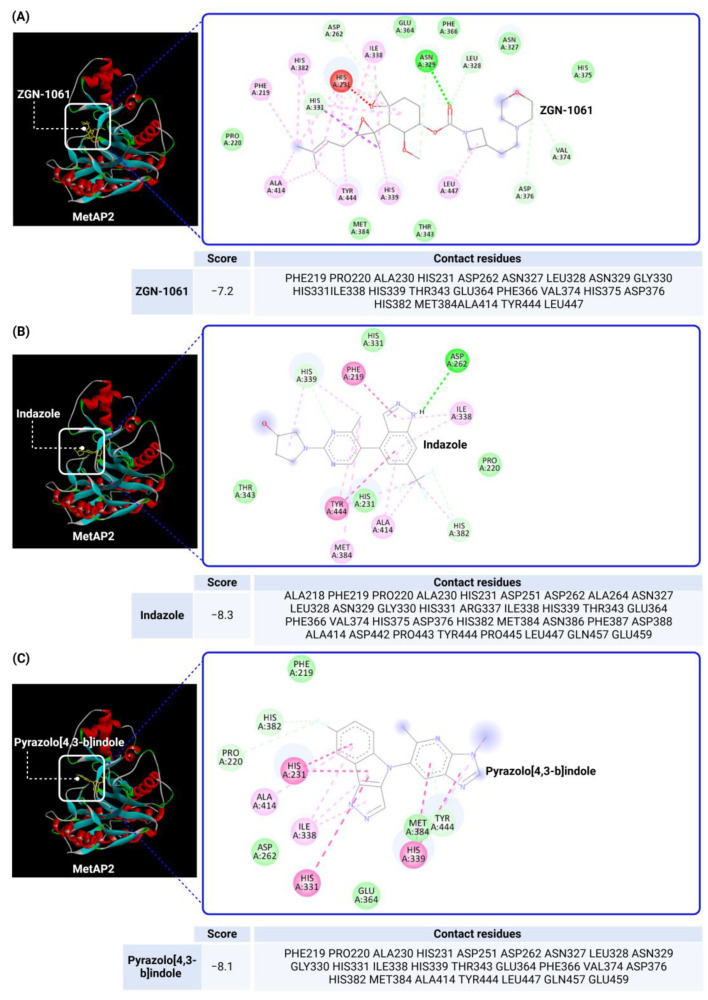
Docking results of ZGN-1061, Indazole, and Pyrazolo[4,3-b]indole with MetAP2. The docking results of ZGN-1061 (**A**), indazole (**B**), and pyrazolo[4,3-b]indole (**C**) with MetAP2 were obtained using CB-Dock2 (https://cadd.labshare.cn/cb-dock2/, accessed on 14 November 2024) and visualized with the Biovia Discovery Studio Visualizer (version 21.1.0.20298). The MetAP2 structure (PDB ID: 1BOA) was obtained from the Protein Data Bank, and ligand structures (SDF files) were retrieved from PubChem. CB-Dock2 was employed to predict the binding sites and calculate binding scores (in kcal/mol), which were −7.2, −8.3, and −8.1 kcal/mol for ZGN-1061, indazole, and pyrazolo[4,3-b]indole, respectively. The 2D interaction diagrams depict the binding interactions with key residues such as His231, His339, Tyr444, and Leu447. Both indazole and pyrazolo[4,3-b]indole formed strong non-covalent interactions with the active site metal ions, while ZGN-1061 demonstrated a balanced binding profile with hydrophobic contacts in the adjacent pocket. The results emphasize the importance of nitrogen-containing warheads and hydrophobic substituents at the 6- and 7-positions for maximizing the binding affinity and stability.

**Table 1 biomolecules-14-01572-t001:** Key Effects and Clinical Trial Stages of MetAP2 Inhibitors.

Inhibitors	Key Effects of the Inhibitor	Clinical Stage	Ref.
**Fumagillin** 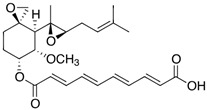	**Weight reduction**: reduced body weight, gonadal (GON), and subcutaneous (SC) fat masses in high-fat diet-fed mice. **P-glycoprotein interaction**: strong interaction energy with P-glycoprotein (Pgp), potentially inhibiting Pgp-mediated drug efflux and enhancing oral drug bioavailability. **Glucose metabolism enhancement**: improved glucose tolerance, enhanced hepatic glucose uptake (NHGU), and increased liver glycogen storage in high-fat and high-fructose-fed dogs.	Preclinical	[57,59,60]
**TNP-470** 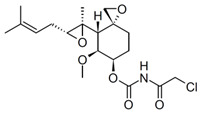	**Weight management**: reduces body weight gain, increases basal metabolic rate, and shifts metabolism from carbohydrates to fat. **Energy balance**: decreases food intake initially and enhances energy expenditure. **T2DM management**: lowers blood glucose and improves glucose tolerance, insulin sensitivity, and pancreatic islet structure; enhances sitagliptin effects.	Preclinical	[63,64,65]
**Beloranib** 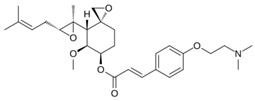	**Weight Reduction**: significant weight loss in dose-dependent manner (0.6, 1.2, and 2.4 mg doses reduced body weight by −5.5, −6.9, and −10.9 kg, respectively, over 12 weeks). **Hypothalamic injury-associated obesity**: effective weight loss in patients with hypothalamic damage (−6.2 kg after 8 weeks) and improved high-sensitivity CRP. **Hyperphagia reduction in Prader–Willi Syndrome**: statistically significant weight loss (−9.5% with 2.4 mg dose) and improved hyperphagia behavior scores. **T2DM**: weight loss (−13.5% with 1.8 mg dose) and HbA1c improvement (−2.0% with 1.8 mg dose), but trial terminated early due to safety concerns (venous thromboembolism).	Clinical Phase II	[73,74,75,76]
**ZGN-1061** 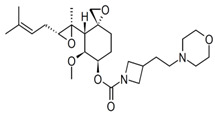	**Weight reduction**: demonstrated significant weight loss in preclinical and early-phase clinical trials targeting obesity. **Improved insulin sensitivity**: enhanced glucose uptake and improved insulin sensitivity in **T2DM** models.	Clinical Phase II	[79]
**Indazole (Compound 38)** 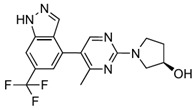	**Weight reduction**: demonstrated robust and sustained dose-dependent body weight loss in obese mouse models over 28 days. **Metabolic improvement**: enhanced insulin sensitivity and glucose metabolism by targeting MetAP2 with high potency and selectivity. **Pharmacokinetics**: showed good oral bioavailability (58%) and stability in human hepatocytes.	Preclinical	[80]
**Pyrazolo[4,3-b]indole (Compound 10)** 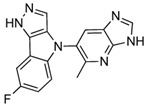	**Weight reduction**: demonstrated dose-dependent body weight loss in DIO mouse models (4% reduction at 30 mg/kg after 7 days). **Metabolic improvement**: maintains good potency and selectivity with acceptable ADME properties (70% oral bioavailability). **Selectivity**: no significant off-target activity observed in a pharmacological profile panel of 100 biological targets.	Preclinical	[81]

**Table 2 biomolecules-14-01572-t002:** Summary of MetAP2 Inhibitors: Binding Analysis, Docking Insights, and Key Results. N.T. indicates not tested.

Inhibitor Type	Binding Analysis Program	Docking Analysis Program	Docking Score	Result	Ref.
**Fumagillin**	Crystallography and LIGPLOT visualization	N.T	N.T	Covalent bond formation with His231 at MetAP-2 active site; involves hydrophobic and water-mediated polar interactions.	[20]
**TNP-470**	Crystallography and LIGPLOT visualization	N.T	N.T	Likely binds similarly to fumagillin at MetAP-2, with modifications at the C6 side chain for efficacy and specificity.	[20]
**Beloranib**	X-ray structural analysis of cocrystals	N.T	N.T	Potent MetAP2 inhibitor with a fumagillol backbone; forms hydrophobic interactions at MetAP2 active site (Leu323, Leu447). Limited clinical development due to poor water solubility.	[68]
**ZGN-1061**	N.T	N.T	N.T	N.T	
**Indazole**	X-ray crystallography, SAR analysis	N.T	N.T	Reversible binding to MetAP2 active site; N2 of indazole coordinates with active site metal ions, with hydrophobic interactions at the 4- and 6-position substituents.	[78]
**Pyrazolo[4,3-b]indole**	X-ray crystallography, SAR analysis	N.T	N.T	N2 of pyrazole warhead coordinates with active site metal ion; 7-fluoro substituent fills a lipophilic pocket, enhancing potency.	[79]

## Data Availability

Not applicable.

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
