# Peer review of "MetAP2 as a Therapeutic Target for Obesity and Type 2 Diabetes: Structural Insights, Mechanistic Roles, and Inhibitor Development"

_biomolecules, 2024, doi:10.3390/biom14121572_

Round 1
Reviewer 1 Report
Comments and Suggestions for Authors
The manuscript biomolecules-3345687 “MetAP2 as a Therapeutic Target for Obesity and Type 2 Diabetes: Structural Insights, Mechanistic Roles, and Inhibitor Development” is a review on the potential role of MetAP2, a methionine aminopeptidase 2 enzyme which involved in the lipid metabolism, energy balance, and protein synthesis.
The work is nicely written and therefore recommended with minor modification as suggested below.
1. Describe the statistics of current diabetic and obese peoples.
2. Specifically mention the two metals bind to MetAP2 at the beginning (page 3, line 106), and explain what the role of the metals are?
3. Cite the reference for 3D structure of MetAP2 in figure 2.
4. Give the chemical structures of inhibitors in Table 1 and 2.
5. Is CB-Dock2 freely accessible?
6. Which software is used in molecular docking study visualization and predicting the interactions?
7. If the figures 3 and 4, were taken from literature than proper permission and citation is required.
8. Delete line 710, page 19. “6. Docking Studies of MetAP2 Inhibitors for Obesity and T2DM”.
Author Response
- Describe the statistics of current diabetic and obese peoples.
⟶ Thank you for your suggestion; I have included relevant global statistics on diabetes and obesity in the introduction.
- Specifically mention the two metals bind to MetAP2 at the beginning (page 3, line 106), and explain what the role of the metals are?
⟶ I have clarified the specific metals binding to MetAP2 and their roles in the catalytic process.
- Cite the reference for 3D structure of MetAP2 in figure 2.
⟶ I have added the citation for the 3D structure of MetAP2 (PDB ID: 1BOA) in Figure 2.
- Give the chemical structures of inhibitors in Table 1 and 2.
⟶The chemical structures have been included in Table 1.
- Is CB-Dock2 freely accessible?
⟶ Yes, CB-Dock2 is freely accessible and is a web-based tool that uses the AutoDock software for molecular docking studies. It provides reliable results by predicting ligand-binding sites with high accuracy and scoring binding poses to facilitate precise docking simulations.
Ref: Liu Y, Yang X, Gan J, Chen S, Xiao ZX, Cao Y. CB-Dock2: improved protein-ligand blind docking by integrating cavity detection, docking and homologous template fitting. Nucleic Acids Res. 2022 Jul 5;50(W1):W159-W164. doi: 10.1093/nar/gkac394.
- Which software is used in molecular docking study visualization and predicting the interactions?
⟶ The following sentence has been added and revised. “The molecular docking study was conducted using CB-Dock2 and visualized with Biovia Discovery Studio Visualizer (version 21.1.0.20298).”
- If the figures 3 and 4, were taken from literature than proper permission and citation is required.
⟶ Thank you for your comment. Figures 3 and 4 were generated as part of the original work conducted in this study and are not taken from any external literature.
- Delete line 710, page 19. “6. Docking Studies of MetAP2 Inhibitors for Obesity and T2DM”.
⟶ It has been deleted.
Reviewer 2 Report
Comments and Suggestions for Authors
The topic is interesting, but I have a few suggestions:
1. The numbers of bibliographic references are doubled. Why ?
2. Regarding the full bibliographic notes, write the font appropriate to the journal name in italics, and the year in bold.
3. Who is compound 10. Compound 10, which is referred to, should be presented line 584
4. I would suggest that the chemical structure of the inhibitors in tables 1 and 2 be presented. lines 601 and 609.
5. Explain the choice of the 6 inhibitors.
Comments on the Quality of English Language
Some chemical terms in the text could be improved.
Author Response
- The numbers of bibliographic references are doubled. Why ?
⟶ It has been revised to comply with the formatting requirements.
- Regarding the full bibliographic notes, write the font appropriate to the journal name in italics, and the year in bold.
⟶ During the process of copying into the journal's manuscript template, the formatting of the bibliographic references appears to have changed. I have revised them to align with the journal's formatting guidelines.
- Who is compound 10. Compound 10, which is referred to, should be presented line 584
⟶Thank you for your comment. I have added a description of Compound 10 and its source in the relevant section to provide clarity.
- I would suggest that the chemical structure of the inhibitors in tables 1 and 2 be presented. lines 601 and 609.
⟶ The chemical structures have been included in Table 1.
- Explain the choice of the 6 inhibitors.
⟶ The following sentence has been added and revised. “Building on this foundation, this review aims to focus on six representative MetAP2 inhibitors with preclinical or clinical research outcomes for type 2 diabetes and obesity, exploring their mechanisms of action, therapeutic potential, and results from these studies.”
Round 2
Reviewer 2 Report
Comments and Suggestions for Authors
I appreciate your response to my suggestions.
You need to re-correct the structures of the last two compounds, namely: Indazole (Compound 38) and Pyrazolo[4,3-b]indole (Compound
10, as they are incorrect.
1. Compound 38 in Table 1 (reference 79) is an R substituent at position 6 of the indazole ring. As written it is not an indazole. This is incorrect.
You need to write this substituent at position 6 on the indazole ring.
2. Compound 10 in Table 1 (reference 80) is not a pyrazolo[4,3-b]indole. It does not correspond to the structural formula of the R radical of compound 10. It is a completely different compound.
It must be re-corrected and written correctly.
Write the R substituent of compound 10 at position 4 of the pyrazolo[4,3-b]indole ring.
Author Response
- Compound 38 in Table 1 (reference 79) is an R substituent at position 6 of the indazole ring. As written it is not an indazole. This is incorrect. You need to write this substituent at position 6 on the indazole ring.
: Thank you for pointing this out. The description has been corrected to accurately represent the substituent at position 6 of the indazole ring.
- Compound 10 in Table 1 (reference 80) is not a pyrazolo[4,3-b]indole. It does not correspond to the structural formula of the R radical of compound 10. It is a completely different compound. : The table has been revised to correctly depict Compound 10 with the R substituent at position 4 of the pyrazolo[4,3-b]indole ring.